# Position: Digital Agents Require Unified Agent-Native Computers

**Yiran Wu** [* 1 2]  **Jiale Liu** [* 1 2]  **Jieyu Zhang** [3]  **Yaolun Zhang** [4]  **Shilong Liu** [5]  **Chi Wang** [6]  **Mengdi Wang** [5]  **Huazheng Wang** [4 2]  **Qingyun Wu** [1 2]

## Abstract

Large language models (LLMs) are increasingly deployed as *digital agents* that perform multi-step digital work on a computer, but the environments in which they operate remain fragmented and task-specific. Our position is that digital agents need AGENT-NATIVE COMPUTER: interaction surfaces that expose system capabilities through compositional observation and action spaces aligned with LLM strengths. To ground this position, we showcase **AgentVM**, an environment running on top of a modern operating system, which integrates Graphical User Interface (GUI)-based and text-based interactions over a shared system state, and factors interaction into modular *environment views*. Through quantitative and qualitative analysis, we show that a unified agent-native computer is essential for building general-purpose digital agents. Code is available at https://github.com/ag2ai/AgentVM/tree/main!

## 1. Introduction

For decades, work in natural language processing has largely focused on defined language understanding or generation problems, evaluated through benchmarks that have treated models as text-in, text-out systems that map a prompt to a response, possibly over a short multi-turn conversation (Wang et al., 2018; 2019; Hendrycks et al., 2020; Liang et al., 2022; Wang, 2022). Recent foundation models, however, demonstrate substantially broader general-purpose capabilities and can operate over long and multimodal contexts (Achiam et al., 2023; Team et al., 2024; Jaech et al., 2024; Anthropic, 2024; Liu et al., 2024a). These capabilities have fueled a

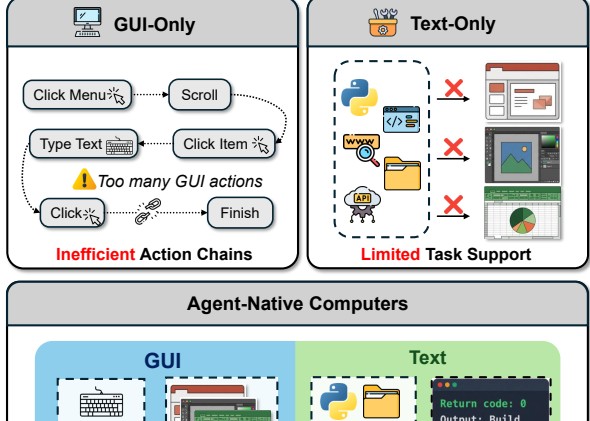

*Figure 1.* Existing agent environments suffer from either inefficient GUI action chains (left) or are limited to API-available tasks (right). Agent-native computers unify both interfaces atop an operating system, providing efficient actions and universal task coverage.

growing interest in using LLMs not only as conversational systems but as *agents* that coordinate tools and perform actions in external environments (Gao et al., 2025; Wang et al., 2024a; Xi et al., 2025; Schick et al., 2023; Wu et al., 2023b; Wang et al., 2023; Nguyen et al., 2025), leading to rapid progress in agent modeling and frameworks (Yao et al., 2022b; Yang et al., 2024a; Wu et al., 2023a; 2024; Hong et al., 2023; Khattab et al., 2023; Liu et al., 2025).

Building agents that can tackle real-world digital tasks has attracted substantial attention, given the potential to materially improve human productivity (e.g., software engineering, web search) and has motivated a growing set of benchmarks and environments spanning diverse task families (Deng et al., 2023; Yao et al., 2022a; Chang et al., 2026). Yet, today's ecosystems remain fragmented and largely task-specific: most environments are limited to a narrow workload and expose either low-level Graphical User Interface control (Xie et al., 2024; Zhou et al., 2023; Niu et al., 2024; Rawles et al., 2024; Shaw et al., 2023) or benchmark-specific, text-based tool APIs (Trivedi et al.,

---

[*]Equal contribution  [1]Pennsylvania State University  [2]AG2AI, Inc.  [3]University of Washington  [4]Oregon State University  [5]Princeton University  [6]Google DeepMind.  Correspondence to: Yiran Wu <ykw5399@psu.edu>, Huazheng Wang <huazheng.wang@oregonstate.edu>, Qingyun Wu <qxw5138@psu.edu>.

*Proceedings of the 43rd International Conference on Machine Learning*, Seoul, South Korea. PMLR 306, 2026. Copyright 2026 by the author(s).

2024; Qin et al., 2023; Yao et al., 2024; Patil et al., 2024; Guo et al., 2024; Luo et al., 2025c; Liu et al., 2024b). This separation creates a critical interface limitation: many realistic end-to-end tasks inherently require both GUI interaction (to navigate, inspect, and verify state) and text-based tools (to transform data or automate edits), and cannot be robustly solved when only one interaction mode is available. As a result, agents are typically engineered for the idiosyncrasies of a particular benchmark environment, and improvements often fail to transfer across task families and platforms.

In this work, we take the position that **digital agents need an AGENT-NATIVE COMPUTER to tackle any digital task**. Such a computer integrates both GUI-based and text-based interaction, with a shared OS state. At the same time, the computer interface can be configured to include different kinds of action and observation spaces. We then build **AgentVM** to implement these design principles. Through experiments and case studies with **AgentVM**, we demonstrate that an AGENT-NATIVE COMPUTER is essential for agents to solve digital tasks efficiently.

## 2. The Rise of Agentic Work

### 2.1. The Rise of LLM Agents

LLM-based agents have quickly become a central paradigm for using LLMs to interact with environments. We use the term *LLM agent* to refer to a system that places an LLM at the core of a perception–decision–action loop: given a goal and contextual information, the model selects actions that drive subsequent computation or interaction (Gao et al., 2025; Wang et al., 2024a; Xi et al., 2025; Sumers et al., 2023; Liu et al., 2023). Rather than producing a single response to a prompt, the agent iteratively observes the consequences of its actions and decides how to proceed through LLM calls.

This formulation has been instantiated across a wide range of applications. In relatively constrained settings, LLM agents augment traditional language tasks by integrating external tools, improving performance on multi-step benchmarks (Yang et al., 2018; Cobbe et al., 2021; Hendrycks et al., 2021; Schick et al., 2023; Wu et al., 2023b). In more complex scenarios, agents serve as controllers for external software and services, coordinating tool use and long-horizon decision making to accomplish open-ended goals (Yao et al., 2022b; Wang et al., 2023; Wu et al., 2023a; 2024). In parallel with these developments, surveys and frameworks increasingly treat LLM agents as a distinct and rapidly evolving line of research (Gao et al., 2025; Wang et al., 2024a; Xi et al., 2025; Luo et al., 2025a; Zhang et al., 2025a).

Industry adoption of LLM agents is accelerating, with products like Manus (Manus, 2026), Claude Cowork (Anthropic, 2026), and Eigent (eigent-ai). They provide agents with ei-

ther cloud-based virtual machines or local desktop systems to assist with digital work ranging from research ideation to file automation. These commercial deployments demonstrate that agents require general-purpose computing environments rather than task-specific interfaces to perform valuable general digital work.

### 2.2. Towards Real-world Tasks

LLM agents are rapidly becoming more capable at solving complex tasks across domains. Thus, more recent benchmarks on LLM agents focus on realistic tasks that require comprehensive abilities such as software engineering (Jimenez et al., 2023; Yang et al., 2024a; Zan et al., 2025), scientific discovery (Boiko et al., 2023; Gottweis et al., 2025; Shojaee et al., 2025), web-based interactions (Zhou et al., 2023; Deng et al., 2023), and threat investigation (Wu et al., 2025; Zhang et al., 2024). These tasks are either directly crawled from the real-world following the exact setting, or are composed to simulate scenarios that can be encountered. *SWE-Bench* (Jimenez et al., 2023) pulls issues from popular GitHub repositories and formulates them as questions. Providing a code execution environment to the agent greatly improves the agent's performance, where the agent can modify code content, execute bash scripts, just like what a software engineer can do. *OSWorld* (Xie et al., 2024) provides an OS system that an agent can interact with through GUI interactions, such as clicking an app icon, or typing with a keyboard. Deep research tasks like *GAIA* (Mialon et al., 2023) requires the agent to perform web search and understand different modality of inputs, and WebArena (Zhou et al., 2023) requires interactions with websites. Recently, *GDPVal* (Patwardhan et al., 2025) introduces digital tasks that have real-world economic values (i.e., tasks that can contribute to a country's GDP). These tasks require comprehensive abilities from agents. To solve the tasks, the agent needs to operate a computer, process multimodal input, and be able to plan and progress in a very long-horizon manner. The required output is also much more complex than an answer: it requires a full written document, or a complete code repository as the end product.

## 3. The Missing Agent Environment

### 3.1. Problem Statement

Humans use computers to solve digital tasks. A *modern computer* runs an operating system that provides services to applications. An *application* is a user-level program that carries out a task by combining its own logic with OS-provided services. A graphical user interface presents state visually and accepts input through pointing and typing, translating interactions into application callbacks that invoke OS services. Applications may use GUIs or text interfaces (e.g., terminal shells), but both rely on the same OS services.

*Table 1.* Required tools and capabilities for different real-world benchmarks. (✓) indicates a tool/capability/feature is required or included. (✗) indicates a tool/capability is not required, but good to have, and is commonly used when agents are built.

| Benchmark | Interaction | | | Modality | | | w/ Env |
|---|---|---|---|---|---|---|---|
| | Code | Search | GUI | Image | Audio | Video | |
| BrowseComp (Wei et al., 2025) | | ✓ | | | | | |
| SWE-bench (Jimenez et al., 2023) | ✓ | | | | | | |
| WebArena (Zhou et al., 2023) | | ✓ | ✓ | ✗ | | | ✓ |
| OSWorld (Xie et al., 2024) | ✗ | | ✓ | ✓ | | | ✓ |
| GAIA (Mialon et al., 2023) | ✗ | ✓ | ✗ | ✓ | ✓ | ✓ | |
| GDPVal (Patwardhan et al., 2025) | ✓ | ✓ | ✓ | ✓ | ✓ | ✓ | |

GUIs support direct manipulation of visible objects, emphasize discoverability over memorization, and align with human perception and control, making them an effective human-centric abstraction of system state and control.

However, graphical interfaces may not be the best interfaces for LLM agents. LLMs are trained primarily on text and reason symbolically. They do not require pixel rendering, and while vision–language models continue to improve, precise graphical understanding and fine-grained control remain challenging. This motivates us to ask the question:

> *What environment does an agent need to complete any given digital task?*

### 3.2. Why Existing Environments Do Not Suffice

The emerging benchmarks and systems described above are built around distinct environments, where each introduces its own state representations, and interaction assumptions, optimized for a particular slice of digital work (Table 1). From the perspective of an agent, this yields a patchwork of incompatible worlds rather than a single, coherent computer on which to learn and act.

**Pure GUI Interfaces.** It seems that the environment from OSWorld (Xie et al., 2024) is already enough: a hosted virtual machine with applications like web browsers, developer tools, and office software. This environment provides a screenshot and an accessibility tree as observations and exposes a suite of GUI-based actions (e.g, click, focus). Any application accessible to humans becomes accessible to the agent. While this environment is general enough, **a pure GUI-based environment can be difficult and inefficient, due to its limited observation space and action space.**

Consider a software engineering task using VSCode. To initialize the environment, GUI agents must go through a long chain of visual actions: click icons, navigate menus, and traverse file pickers. Once the code is finally visible,

the process reveals a critical inefficiency. The interface renders native text files into pixels within a screenshot, forcing the agent to visually encode the image back into language tokens. This creates a redundant conversion loop from text to image and back to text, whereas feeding the raw code directly to the model would remain lossless and immediate.

**Pure Text-Based Interfaces.** In comparison, many existing interfaces are text-based. SWE-agent (Yang et al., 2024a) proposes the idea of agent computer interface, and designs pure text-based actions such as running bash scripts for SWE tasks. In order to solve tasks from GAIA (Mialon et al., 2023), agents use dedicated text-based tools including python interpreter, web search and visiting, image & video analysis (Fourney et al., 2024; Song et al., 2024; Roucher et al., 2025; Qiu et al., 2025; Zhang et al., 2025b). These text-based actions are tailored for LLM-based agents, which significantly improves agents' performance in agentic tasks like software engineering and deep research. Still, pure text-based interfaces have their limitations. **Many digital tasks require interactions with GUI-based applications**, such as Microsoft PowerPoint and Word. For example, to create a PowerPoint presentation, GUI interactions are essential, and visual verification is required. Another case is the interaction with web applications. Since deep research involves primarily information gathering (Zeng et al., 2025; Wei et al., 2025), text-based web browsing is useful in most cases. However, web applications can block automated access or require complex interactions that are mostly GUI-based, which limits the utility of text-based browsing.

### 3.3. Proposed Solution: Unified Agent-Native Computer

Existing environments use either GUI-based or text-based interaction. GUI-based environments offer generality but require costly modality conversion and multi-step navigation. Text-based environments align with LLM strengths through structured interaction, but are limited to applications with explicit APIs or tooling. These modalities remain scattered across benchmarks with little overlap, and neither

alone suffices for general-purpose digital work. Prior work indicates that structured, semantically rich interfaces tailored for agents can improve the reliability and capability of agents (Yang et al., 2024a;b). Rather than repurposing human GUIs, we therefore propose building an AGENT-NATIVE COMPUTER, which exposes the same capabilities through symbolic and multimodal representations.

## 4. Agent-Native Computer: A Conceptual Framework

We propose a new conceptual framework, AGENT-NATIVE COMPUTER, which is a unified environment that can be used by agents to tackle arbitrary digital tasks, offering these key advantages:

- **Integrated GUI-based and text-based interaction surfaces.** The computer provides the standard GUI interface of an OS and augments it with text-based action interfaces that can be mounted alongside it. These interfaces provide structured, semantic operations, such as file manipulation and web retrieval, that complement low-level GUI events. Rather than treating GUI and text interaction as separate paradigms, we treat them as coequal mechanisms for invoking the same underlying OS abstractions.

- **Shared OS state across all interaction interfaces.** All interfaces operate on the same underlying sandboxed OS state $S$. Any change made through one interface is consistently reflected in the others. For example, when a web tool downloads a file into the `Downloads` directory, that file becomes part of the OS file system and can be opened through a GUI application. Because every interface changes the same OS state, the results of an action are immediately visible across all other interfaces.

- **Environment Views as factorizations of interaction.** The actions and observation spaces of the environment are compositional, and can be grouped to form different *Environment Views*. Each view exposes the *same* latent OS state through a different factorization of observations and actions: one view may emphasize window hierarchies and GUI controls; another may expose structured file metadata and editor operations. These views differ only in how they group the observations and actions over the same environment, allowing different interaction patterns to be expressed while remaining grounded in a single underlying environment. This makes it easy to configure the environment for different needs, or to create agentic systems where each agent can be dedicated to its assigned environment view.

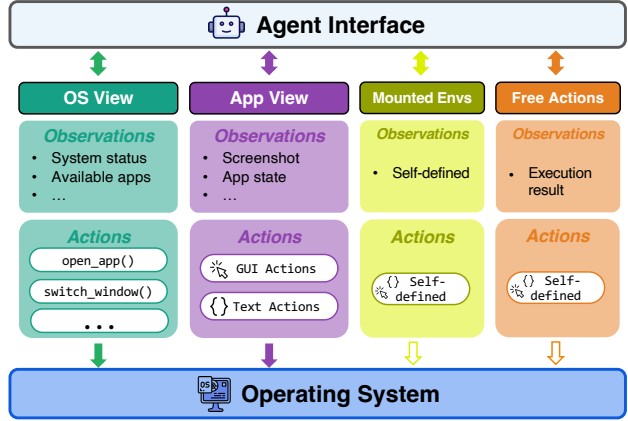

*Figure 2.* Illustration of AGENT-NATIVE COMPUTER

### 4.1. Preliminary

We now formalize the environment in more precise terms. We begin by introducing a POMDP-based formulation as the underlying interaction model.

**POMDP as underlying problem formulation.** The interaction between an agent and its environment can be framed as a Partially Observable Markov Decision Process (POMDP) (Chen et al., 2024; Zhao et al., 2024a; Kaelbling et al., 1998). Formally, a POMDP is defined as a tuple $(\mathcal{U}, S, \mathcal{A}, \mathcal{O}, T, R)$, where $\mathcal{U}$ denotes the space of all possible tasks (e.g., user instructions or goals), $S$ is the latent state space underlying the environment, $\mathcal{A}$ is the set of permissible actions (often represented as textual commands or API calls), $\mathcal{O}$ is the observation space containing textual or multimodal feedback, $T : S \times \mathcal{A} \to \mathcal{P}(S)$ is the state transition function mapping a state-action pair to a probability distribution over possible next states, and $R : S \times \mathcal{A} \to \{0, 1\}$ is the reward function that indicates whether a given action at a particular state leads to eventual task success or failure.

To describe our environment, we focus mainly on the action space $\mathcal{A}$ and the observation space $\mathcal{O}$. We note that $\mathcal{U}$ and $R$ are not used here. Although the state space $S$ and transition dynamics $T$ are always present in an environment, in our setting they cannot be specified comprehensively or accurately, and will therefore remain implicit.

### 4.2. Environment View

The operating system is a complex environment whose complete state ($S$) we can never fully enumerate. Instead, we can only take partial observations and act based on them. This is also desirable in practice; we do not need full observability to choose an action, and actions typically affect only a localized portion of the overall state, for which the current observation provides sufficient coverage. *Thus, a given observation space and action space are usually grouped*

*together and self-contained.* To formalize this notion of a localized, partial interaction with the environment, we introduce the concept of an Environment View.

**Definition 1** (Environment View). *An **Environment View** $e_i$ is a tuple $(\mathcal{O}^{e_i}, \mathcal{A}^{e_i})$, consisting of an observation space and an action space. The view $e_i$ can be interpreted as an abstraction of the underlying environment; it provides a task-relevant representation of the full system state.*

The GUI interface defined in OSWorld (Xie et al., 2024) can be seen as one type of Environment View of the operating system. Its observation space is the screenshot (or accessibility tree), which captures the current screen of the OS environment. Correspondingly, the defined actions (e.g., click) both depend on the current screen and operate on that screen, leading to changes that are reflected in subsequent screen observations.

Beyond the screen view of the OS, which is a human-centric interface to applications, we consider how to construct more agent-native environment views from the OS. At a high level, the computer provides a basic interface for users to navigate and open different applications, and each application exposes its own specific interface. This generic OS design suggests a two-layer structure: a top-level OS layer and an application layer from which we can build agent-native observation and action spaces.

**The OS-level Environment View $e_{os}$**  There should be an OS-level view of the computer, which can be used to manage the operating system. For example, useful observations include available applications, and active windows opened in the computer, grouped with actions that manipulate them, including open or close applications / windows.

**The App-level Environment View $e_{\{app\}}$**  We view any opened application as a "sub-environment", where app-specific observations and actions can be added. Note that an application can open multiple files with several windows, where each window can be further split into separate environment views if needed. By default, an application view $e_a$ will take the screenshot of the opened application (in full window) as the observation and standard GUI actions as the action space. Based on this, additional observations and actions can be created and integrated into the view (Jia et al., 2025). For example, for a web browser, an observation to list out all opened tabs in sequence can be included, with a bit of content from each website. Correspondingly, we can create a corresponding action "switch_tab" that LLM can use to switch to any tab. We note that such an action should be treated as an advanced GUI action instead of a text action, since it is used to manipulate the GUI interface. The described observation and action can also be realized through screenshots and GUI actions, but the described text-based observation and action are much more efficient and

easier to use for LLM agents. Based on this framing, more observations and actions can be added, making it easier for LLM agents to take actions, and making the agents less dependent on the raw screenshot and GUI actions.

*Remark* 1 (Hierarchical OS-APP Views). Conceptually, the OS and App views are hierarchically structured: Using the OS View, we can control the applications in the system; meanwhile, the App views are specific to each application.

*Remark* 2 (Dynamic Environment Views). We note that the views can also be dynamic: when applications are opened or closed, the corresponding app views will also be mounted or removed. Other rules can also be crafted to control what observation / actions the agent can see for more grounded and precise operations.

### 4.3. Mounting Additional Environments and Actions

In addition to OS-derived views, AGENT-NATIVE COMPUTER can mount *OS-independent*, self-contained environments as well as expose actions that are not tied to any particular environment. To distinguish these capabilities, we classify actions into two types:

**Definition 2** (Context-Bound Action). *A Context-Bound Action $a^{e_i} \in \mathcal{A}^{e_i}$ is an action associated with a specific environment (view) $e_i$. When executed, $a^{e_i}$ is applied within $e_i$. Such actions are* state-sensitive*: their effects may depend on the current state of $e_i$, and executing $a^{e_i}$ may in turn modify the state of $e_i$.*

**Definition 3** (Free-Form Action). *A Free-Form Action $a^i$ is not associated with any environment and can be executed without relying on an environment state and functions as a direct (stateless) function call.*

Under this design, AGENT-NATIVE COMPUTER supports mounting additional environments with their own consistent observation and action interfaces (e.g., a text-based web browser implemented via Playwright), as well as registering free-form actions (e.g., an API call to a web service).

### 4.4. Overview

We now summarize the design to provide an overview. In the same virtual machine, several types of views can exist:

- The OS-level view $e_{os}$: defined system-level status as observation and actions as control.

- The App-level view $e_{\{app\}}$: screenshot of the app and other defined app-level status as observation, GUI, and defined text-based actions.

- Mounted environments $e_{\{ext\}}$: Here the defined observation space and action space can be arbitrary; they may be dependent on the virtual machine or not.

- Mounted free-form actions $a^j$: stateless actions that do not depend on any environment.

All of the above can be "flattened" and grouped together, where the complete view is an aggregation of all current accessible groups ($\mathcal{O} = \bigcup_i \{o_{e_i}\}, \mathcal{A} = \bigcup_i \{a^{e_i}|a^i\}$). At the same time, different observations and actions can be arbitrarily grouped into one environment view flexibly, for use by an agent. Such a design makes the computer fully configurable and well-suited to build agentic systems, where specialized agents will be assigned required environment views.

## 5. Experiments and Analysis

As a concrete step toward realizing this vision, we implement **AgentVM** —a unified environment that instantiates the core design principles of AGENT-NATIVE COMPUTER.

### 5.1. Initial Implementation

We take the OS environment from OSWorld (Xie et al., 2024) as the starting point for building. Several recent works explore related directions in making operating systems more amenable to agents (Wang et al., 2025b; Trivedi et al., 2024). For example, Song et al. (2025) introduced *coding agent* equipped with the ability to run bash scripts to solve OSWorld tasks better, while Jia et al. (2025); Wang et al. (2025b) design *text-based GUI actions* that reduce reliance on raw GUI interactions, thereby improving both performance and efficiency on OSWorld benchmarks. **AgentVM** naturally integrates all of them, while offering easy customization of more observations and actions. More implementation details are in Appendix D. On top of **AgentVM**, we conduct experiments on different datasets to confirm the effectiveness of the proposed framework.

### 5.2. OSWorld Results

OSWorld is a GUI-based benchmark where agents complete tasks by controlling a computer through low-level GUI operations. We select "workflow" subset of OSWorld for benchmarking, since we want to target a more complex scenario where the agents need to coordinate across multiple applications under partial observability.

**Experimental Design**   Our goal is not to introduce a novel agent architecture, but to validate the design principles of AGENT-NATIVE COMPUTER. We hold the agent structure fixed and vary only the interaction surfaces exposed by the environment. We build on a hierarchical agent framework inspired by Agent S2.5 (Agashe et al., 2025), consisting of a planner agent that interprets screenshots and generates natural language action plans, and a grounding agent that translates plans into executable OS actions. Starting from this GUI-only baseline, we progressively augment the agent with the proposed design principles:

- **Tools**: We mount text-based tools via MCP (Model Context Protocol), ported from OSWorld-MCP (Jia et al., 2025). The agent can choose between invoking structured tool calls or performing operations through GUI actions. This introduces the first layer of text-based interaction alongside the existing GUI interface.

- **OS Interface**: We add the OS-level environment view, exposing system state through structured observations (window status, application info) and text-based OS actions (open_app, switch_window, etc). This provides the agent with explicit, symbolic access to OS-level information that would otherwise require multi-step GUI navigation to discover, or demand that the agent recall and reason over a long interaction history.

- **Coding**: Finally, we enable both python and bash code execution for the agent. The agent can delegate tasks to a coding agent that executes them. This allows the agent to perform computation and file manipulation programmatically rather than through GUI interactions.

*Table 2.* Comparison of agent performance on OSWorld with different interfaces.

| Setting | Tools | OS View | Code | Result |
|---|---|---|---|---|
| Baseline | | | | 41.54 |
| + Tools | ✓ | | | 46.72 |
| + OS View | ✓ | ✓ | | 47.87 |
| + Code | ✓ | ✓ | ✓ | **51.30** |

**Analysis**   Table 2 shows that **performance improves monotonically as we layer additional agent-facing interfaces onto the base GUI environment.** The largest gain (+5.18 points) comes from mounted tools, which provide text-based alternatives to lengthy GUI action chains. This suggests that workflow failures often stem from brittle multi-step GUI sequences required for simple operations–text-based tools allow direct execution of these intents (Gou et al., 2025; Gonzalez-Pumariega et al., 2025).

The OS Interface and Coding layers provide complementary benefits. OS-level views expose system state symbolically, eliminating repeated screenshot parsing and navigation. Code execution enables reliable batch processing and file operations that would otherwise require fragile GUI steps. Critically, these additions do not replace GUI interaction but expand the action repertoire over the same OS state. The monotonic gains confirm that unifying GUI and text-based interaction surfaces reduces grounding overhead and improves agent reliability on real-world digital tasks.

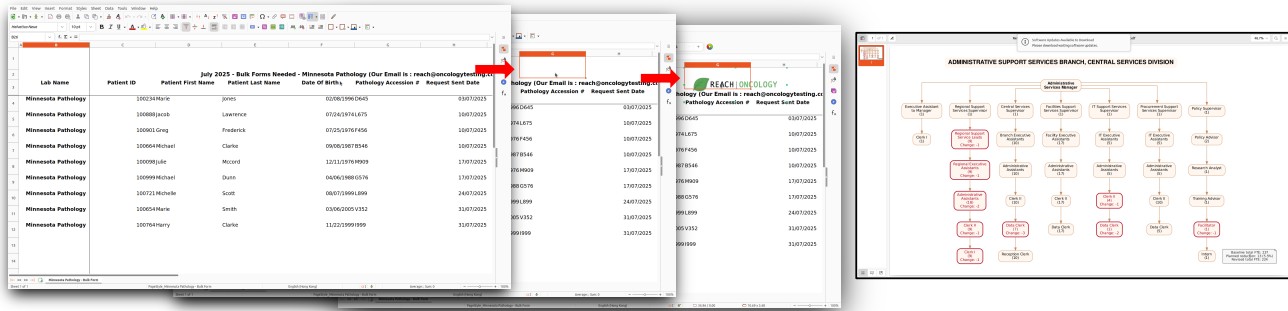

*Figure 3.* (**Left**) Case Study 1. A logo is copy pasted into head of Excel file. (**Right**) Case Study 2. The updated organization chart.

*Table 3.* Two case study examples from GDPVal(abbreviated; full text and execution trajectory in Appendix E).

**Case 1: Medical Secretary**

**Goal:** Prepare lab-specific tissue-request bulk forms and email templates for an oncology testing center.

**Given:** A reference Excel sheet of patient tissue requests for three pathology labs.

**Deliverables:** (i) Three Excel bulk-form workbooks (one per lab) including Reach Oncology's logo and email, with additional columns and rows sorted by request date; (ii) three Word email templates (one per lab) with lab-specific subject/greeting requesting status updates for the attached form.

**Case 2: Administrative Services Manager**

**Goal:** Prepare an information package proposing a minimum 4% FTE reduction for a branch.

**Given:** Budget Planning Principles (Word), current org chart (PDF), current FTE report (Excel).

**Deliverables:** (i) Revised branch organizational chart (PDF); (ii) updated FTE reduction report (Excel); (iii) briefing note (Word) covering background, proposed reductions, and alignment to the budget principles.

**Case 3: Maze Puzzle**

**Goal:** Determine whether a path exists that visits all designated cells exactly once and returns to the starting position.

**Given:** An Excel spreadsheet in which each cell represents a plot of land, with cell colors indicating ownership.

**Deliverables:** A Yes/No answer indicating the existence of a valid closed traversal path.

## 5.3. Case Study

We present two cases from GDPVal and one case from GAIA to illustrate why a *unified* agent virtual machine is important (Table 3). We test the agent with different execution plans to understand how the agent can solve the tasks. We observe a recurring pattern across tasks: GUI interaction is most valuable at *task initialization and verification*: to inspect the given files and ground the agent's understanding of what the workspace looks like, and to visually verify that deliverables satisfy formatting and presentation requirements. In contrast, text-based actions are essential to complete the tasks since LLMs excel at using them.

**Setup.** We build an agent based on GPT-5.2 with access to (i) GUI actions (e.g., click, drag) and (ii) text-based actions (e.g., `execute_bash`). The environment returns either screenshots (for GUI actions) or textual outputs (for text-based actions). For GUI actions, the agent specifies the target element in natural language; we then dispatch the action to the OpenAI Computer Use Agent (CUA) model to localize the pixels and execute the interaction.

**Case 1: Medical Secretary.** In this task, the agent assumes the role of a medical secretary at an oncology testing center. Given a reference Excel spreadsheet, it must (1) split patient rows into three lab-specific bulk forms and (2) draft three corresponding email templates. When restricted to a GUI-only action/observation space, the agent requires too many steps and fails to complete the task reliably. With the unified environment, the agent primarily uses `run_python` to parse the spreadsheet and generate the three output files, enabling completion of the deliverables. However, an important requirement is to preserve the original spreadsheet's formatting and embedded assets (notably the center's logo). Purely programmatic operations on spreadsheets do not reliably preserve these elements.

Based on these observations, we design the following agent workflow: (1) use GUI actions to understand the original file's visual structure; (2) duplicate the original spreadsheet three times; (3) apply programmatic edits to the copies for table updates; and (4) use GUI to verify that formatting and logos are intact and correct as needed. **This combination of programmatic editing plus visual verification is critical**: The agent accurately executes edits via Python but often removes the logo (emitting a warning that images are omitted), then identifies the issue through screenshot examination, and reattaches the logo via GUI actions.

**Case 2: Administrative Manager.** The agent works as an administrative manager responsible for proposing an FTE reduction plan. The task requires integrating information from multiple documents (budget guidelines, an organizational chart, and an FTE roster). Without a human-written plan, the

agent mostly uses the text-based actions to examine the files: the `file_reader` tool to read PDFs, and `run_python` to read the Excel file. While other deliverables can be created as required, the agent fails to produce the revised chart: the agent uses LibreOffice Impress (GUI) to open and edit based on the original chart, but becomes stuck during GUI edits. Instead, we find the following workflow produces better results: (1) Use GUI to open and inspect the organizational chart and to understand the reporting structure, which is difficult to infer from raw text alone; (2) generate a revised chart programmatically through Python. The agent then produces a revised organizational chart preserving the original formatting.

**Case 3: Maze Puzzle.** We also test on a task from GAIA (Mialon et al., 2023) to illustrate the limitations of traditional text-based agents and how our interface provides a more effective interaction paradigm. The task requires the agent to solve a maze puzzle represented by colored cells in an Excel spreadsheet. Previous GAIA-focused agents typically lack GUI capabilities (Qiu et al., 2025; Su et al., 2025), using text-based `file_reader` to read the Excel file and `run_python` to analyze the color distribution. However, reconstructing the spatial layout of the maze from raw cell data proves extremely challenging, which takes more than 5 steps to handle. In contrast, with our OS interface, the agent opens the file in its native application, rendering the solution immediately apparent via screenshot.

## 6. Alternative Views

### 6.1. View 1: GUIs Alone Are Sufficient

An alternative perspective argues that **agent-specific interface layers are just a temporary scaffold that will become unnecessary as vision-language models improve.** This view argues that scaling model training will close the remaining performance gap. If agents achieve human-level GUI proficiency, GUIs become sufficient, making text-based abstractions just add unnecessary overhead.

We argue that this GUI-only perspective confuses capability with optimality and overlooks fundamental inefficiencies in pixel-based interaction for language models. First, GUI operations require long action chains where errors compound. Even with perfect visual grounding, simple operations demand many sequential steps. Each step introduces potential failures in visual parsing or action execution. SWE-Agent demonstrates that structured text interfaces dramatically reduce action chain length and improve reliability (Yang et al., 2024a). The performance gains in Table 2 when adding text-based interfaces confirm that this is not a capability gap but a structural inefficiency in GUI-based interaction.

Secondly, economically valuable tasks inherently have different optimal interaction modalities regardless of agent

capability. Our case studies in Sec. 5.3 illustrate this empirically: successful agents use Python scripts for bulk spreadsheet edits but switch to GUI for visual inspection of formatting. Pure GUI agent must click through hundreds of cells individually even when applying identical operations. This is suboptimal by design, not by capability limitation. The goal should be unifying GUI and text-based interfaces so agents select the appropriate modality for each subtask, precisely the design philosophy embodied in **AgentVM**.

Overall, the question is not whether agents can master GUIs, but whether GUIs alone suffice for digital work. Our results demonstrate that GUI-only faces structural inefficiencies from error-prone action chains and task-modality mismatches that persist regardless of model capability.

### 6.2. View 2: APIs Alone Are Sufficient

A natural objection holds that **text-based actions alone suffice for digital agents once APIs and wrappers are comprehensively developed.** Under this view, GUI interaction is merely a temporary gap in tooling rather than a fundamental requirement.

We defer detailed discussion to Appendix A and briefly respond here: The objection validates our framework rather than undermining it. Proprietary applications lack programmatic interfaces, accessibility trees provide incomplete representations, and many tasks inherently require visual verification—barriers that persist regardless of tooling investment. AGENT-NATIVE COMPUTER addresses these limitations by treating textual actions and GUIs co-equally.

## Conclusion

This paper positions that digital agents require purpose-built environments fundamentally different from those designed for human interaction. As LLM-based agents increasingly tackle economically valuable digital work, fragmented landscape of task-specific benchmarks and single-modality environments has become a critical bottleneck. We propose the concept of Agent-Native Computer: a unified environment that exposes system capabilities through interaction surfaces aligned with LLM strengths. As a concrete step towards the vision, we developed an implementation that realizes these principles on top of a modern OS. Experiments on OSWorld demonstrate that progressively adding framework elements yields monotonic performance improvements, while case studies on complex tasks illustrate why both modalities are essential: GUI interaction enables inspection and visual verification; text-based actions provide efficient bulk operations. Neither alone suffices for real-world digital work.

As LLM agents deploy to real-world digital tasks, their interaction environments must evolve accordingly. While **AgentVM** demonstrates that agent-native environments

can wrap existing systems, we anticipate that widespread agent deployment will eventually drive deeper architectural changes, with applications exposing rich, structured interaction surfaces by design rather than through retrofitting. **Just as GUIs emerged to match human perceptual capabilities, agent-native interfaces may become first-class citizens in tomorrow's software.** Our work provides both a conceptual framework for this future and empirical evidence that unified interaction surfaces are essential for general-purpose agents.

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

## A. Expanded Discussion on Alternative View 2

This view states that **text-based actions alone are sufficient for digital agents, provided that APIs and text-based wrappers are sufficiently developed.** Under this view, the need for GUI interaction is a temporary artifact of incomplete tooling rather than a fundamental requirement. Recent developments lend credibility to this position: Playwright-based MCP servers extract web page state as structured accessibility trees (Microsoft, 2024), browser-use frameworks translate LLM outputs into automated browser actions (browser-use contributors, 2024), and tools like SWE-agent demonstrate that software engineering tasks can be accomplished through purely text-based interfaces (Yang et al., 2024a). If such wrappers could be systematically developed for all applications, a purely text-based environment might suffice, and the complexity of integrating GUI-based and text-based interaction surfaces would be unnecessary.

We argue that this view underestimates fundamental barriers that make complete text-based coverage infeasible in practice. While text-based tools have proven highly effective for specific domains, they cannot generalize to arbitrary digital work. Below we examine three categories of obstacles: proprietary constraints, representational limitations, and tasks requiring visual verification.

**Proprietary applications lack programmatic interfaces.**
Most commercial applications do not expose comprehensive programmatic interfaces, and their developers have little incentive to create them. Microsoft explicitly states that it "does not currently recommend, and does not support, Automation of Microsoft Office applications from any unattended, non-interactive client application"[1]. Office applications assume an interactive desktop with modal dialogs, user identity, and licensing mechanisms that cannot be bypassed programmatically. What's more, it is often deliberately restricted when programmatic access exists. Unlike LibreOffice and other open-source alternatives used in our AgentVM implementation, the vast majority of commercial applications will never expose the interfaces necessary for pure text-based automation. Companies benefit commercially from user engagement with their GUIs and face no market pressure to enable automation that could reduce this engagement or empower competitors.

**Accessibility trees provide incomplete representations.**
Text-based approaches often rely on accessibility trees or DOM structures, which have fundamental representational gaps (Yang et al., 2025; Gou et al., 2025; Luo et al., 2025b). The accessibility tree is designed to support assistive technologies like screen readers, capturing semantic structure but not visual presentation. Dynamic user interfaces with hover menus, or runtime-loaded components disrupt snapshot accuracy, leading to incomplete or incorrect state representations (Fernandes et al., 2011; Bajammal & Mesbah, 2021). Elements without proper accessibility labels (images, icons, SVGs lacking alt text or ARIA attributes) become invisible to text-based tools. In reality, many web pages are often poorly structured, with broken accessibility semantics, custom tags, and non-standard interactive patterns. To build agents that match human behavior, we need more than just a DOM-only signal.

**Many tasks require visual verification.** Certain digital tasks inherently demand visual judgment that cannot be reduced to text. For example, creating a PowerPoint presentation requires not only adding content but verifying that slides are visually balanced, that charts render correctly, and that formatting appears professional. The agent must see the result of its actions to confirm success. Image editing,

---

[1]Link here.

document understanding, graphic design, and many more tasks share this requirement (Zhang et al., 2025c; Zhao et al., 2024b; Hu et al., 2023; Deng et al., 2025; Bi et al., 2025; Liu et al., 2026). Even for tasks that seem purely informational, visual verification serves as an error-correction mechanism: if a GUI action fails silently or produces unexpected results, the agent can detect this only through visual inspection. A pure text-based environment would leave agents unable to verify outcomes for a substantial class of real-world digital work.

**Our Response** The objection validates rather than undermines our position: the barriers to pure text-based automation are exactly why AGENT-NATIVE COMPUTER integrates both modalities. The key insight is that sufficiency cannot be achieved through any single interaction surface. AGENT-NATIVE COMPUTER treats text-based and GUI-based interaction as complementary Environment Views over shared state, both integral for general-purpose digital work.

## B. More Discussions on Future Directions

### B.1. Scaling Agentic Work Through Parallel Execution

A significant advantage of the environment view abstraction proposed in Sec. 4.2 is its natural support for parallel agent execution. By factoring the environment into independent views, each operating over shared OS state, AGENT-NATIVE COMPUTER enables architectures where specialized agents work concurrently on isolated subtasks. This pattern has proven effective in state-of-the-art agentic systems, suggesting that parallelization through modular environment decomposition may be essential for scaling agents to complex tasks (Gottweis et al., 2025; Novikov et al., 2025; Manus, 2025).

Google's AI Co-Scientist exemplifies this approach in the scientific domain. The system employs a coalition of specialized agents that operate asynchronously to generate and refine research hypotheses (Gottweis et al., 2025). The asynchronous task execution framework enables the system to allocate computational resources dynamically, with each agent operating in its own execution branch while contributing to a shared knowledge pool. The AI Co-Scientist can accomplish in days what previously required weeks of human effort, demonstrating the practical benefits of parallel agent architectures. Similar parallel designs appear in other domains: AlphaEvolve orchestrates an ensemble of LLMs that propose program variants concurrently (Novikov et al., 2025), Manus Wide Research deploys over 100 general-purpose agents in parallel to tackle high-volume research tasks, with each sub-agent running on a dedicated virtual machine to avoid context window degradation (Manus, 2025).

### B.2. Safety Considerations in Unified Agent Environments

The hybrid nature of AGENT-NATIVE COMPUTER raises unique safety challenges that neither pure-GUI nor pure-text environments face in isolation. When agents have simultaneous access to visual interaction and code execution, the attack surface expands considerably. A malicious prompt injection could exploit one modality to compromise another, for instance using GUI actions to navigate to a page containing adversarial content that triggers harmful shell commands.

Sandboxing provides the primary defense by isolating agent execution from host systems, typically through virtual machines or containers with restricted resource access and rollback capabilities. AGENT-NATIVE COMPUTER adopts this architecture by running all agent interactions within a sandboxed VM while the framework orchestrates actions across environment views from outside the isolation boundary, ensuring that file system changes and network requests remain contained. This approach aligns with emerging institutional efforts: the UK AI Safety Institute's Inspect Sandboxing Toolkit implements similar separation of inference and execution with tiered isolation options (AI Security Institute), Google Cloud's GKE Agent Sandbox provides Kubernetes-native isolation with pod snapshots for state restoration.

Beyond containment, hybrid environments introduce challenges for safety evaluation and runtime monitoring. Existing benchmarks typically focus on single modalities, such as OS-harm for GUI agents (Kuntz et al., 2025) or Agent-SafetyBench for tool-using agents (Zhang et al., 2024), leaving cross-modality attack vectors underexplored. OpenAgentSafety provides a step toward comprehensive evaluation, finding that more than 50% of safety-vulnerable tasks result in unsafe actions even for frontier models (Vijayvargiya et al., 2025). Furthermore, verification mechanisms in unified environments must track state changes across GUI, text, and code execution simultaneously to detect anomalous behavior patterns that span interaction modalities. AGENT-NATIVE COMPUTER could serve as a platform for developing hybrid safety benchmarks and runtime monitors, helping to identify failure modes unique to unified environments before deployment.

## C. Related Work

### C.1. Agentic Work Benchmarks

Here we provide a broader survey into the landscape of benchmarks related to agentic work.

**Web Benchmarks**   Benchmarks for web agents have evolved from simple element interaction (Shi et al., 2017; Yao et al., 2022a) to handling complex, visually rich environments. WebArena (Zhou et al., 2023) established a baseline by providing a self-hostable environment simulating platforms, which VisualWebArena (Koh et al., 2024) later extended to include visually demanding tasks. To address the limitations of static environments, WebVoyager (He et al., 2024) and Online-Mind2Web (Xue et al., 2025) introduced live execution protocols.

**Enterprise Workflow**   A line of work specifically target domain-specific corporate automation. WorkArena (Drouin et al., 2024) targets corporate productivity by simulating enterprise workflows. The Agent Company (Xu et al., 2024) simulates a complete software development firm where agents need to complete long-horizon tasks across multiple simulated apps. WorkBench (Styles et al., 2024) tests agent's capability in realistic administrative tasks.

Complementing these broad simulations, other datasets isolate specific professional skills and policy adherence. $\tau$-bench (Yao et al., 2024) and $\tau^2$-bench (Barres et al., 2025) evaluates agents in regulated industries like retail and airlines, measuring their ability to strictly follow complex policy guidelines while interacting with users. SheetRM (Chen et al., 2025) benchmarks advanced spreadsheet reasoning, testing an agent's ability to handle ambiguous data processing tasks. OfficeBench (Wang et al., 2024b) and OfficeQA (The Mosaic Research Team, 2025) further round out this category by focusing on multi-app coordination and grounded reasoning over proprietary enterprise documents.

**Deep Research**   Deep Research benchmarks target the high-level reasoning and information synthesis required for digital work. GAIA (Mialon et al., 2023) and AssistantBench (Yoran et al., 2024) evaluate general assistant capabilities through unambiguous but rigorous questions and time-consuming administrative tasks. Odyssey-Bench (Wang et al., 2025a) extends this into long-horizon productivity, requiring multi-app coordination across tools like Excel and Word over simulated days. Complementing these, Bamboogle (Press et al., 2023) and BamTwoogle (Aksitov et al., 2023) specifically target non-Googleable questions, forcing agents to perform multi-hop reasoning across disparate sources rather than relying on direct retrieval. DeepResearch-Bench (Du et al., 2025) and DeepSynth (Zhang et al., 2026) focus on the quality of output in professional research contexts. The former consists of PhD-level tasks designed to evaluate citation accuracy and report quality across diverse fields, while the latter isolates the post-retrieval synthesis capability using oracle contexts from gold-standard survey papers. Together, these benchmarks shift the evaluation metric from successful step execution to the factual accuracy and coherence of the final work product.

**Computer Use**  Significant efforts have been made to simulate full desktop and command-line environments. OS-World (Xie et al., 2024) provide scalable, multimodal environments for Ubuntu and other platforms, enabling agents to interact with the OS via code or GUI. WindowsAgentArena (Bonatti et al., 2024) adapts the desktop paradigm specifically for Windows 11, utilizing Azure for parallel evaluation. For more technical digital work, Terminal-Bench (Merrill et al., 2026) evaluates agents on complex command-line tasks in a sandboxed Linux environment, moving evaluation closer to developer workflows.

In the mobile domain, AndroidWorld (Rawles et al., 2024) and Mobile-Env (Zhang et al., 2023) offer dynamic environments for reproducible task generation, with the latter utilizing ADB-based interaction for WikiHow tasks. These are complemented by large-scale datasets like Android in the Wild (Rawles et al., 2023), which captures human demonstrations, and AndroidLab (Xu et al., 2025b), which focuses on visual grounding for VLMs. Bridging the gap between these distinct ecosystems, Crab (Xu et al., 2025a) introduces an evaluation framework designed for cross-environment tasks, testing an agent's ability to coordinate actions between desktop and mobile interfaces.

## D. OSWorld Implementation Details

**Agent Implementation**  Following Agent S2.5 (Agashe et al., 2025), the agent we developed has a hierarchical structure. When given a user task, a planner agent takes it along with the current screenshot, then generates a natural language plan describing the next action to take. This plan includes specific action calls with natural language descriptions of target elements, such as agent.click("the File menu button in the top toolbar"). Then, a grounding agent processes these natural language descriptions by analyzing the current screenshot to identify the described elements and translate them into precise pixel coordinates. For visual elements, a VLM locates the target based on the description, while text-based targets use OCR to identify word positions on screen. Once coordinates are determined, the action is converted to executable PyAutoGUI commands and performed. After that, a reflection agent analyzes this trajectory, examining whether previous steps achieved their intended effects and providing feedback to guide subsequent decisions.

The agent maintains procedural memory by tracking the execution trajectory across multiple turns. Each turn stores the generated plan, the executed action code, and the corresponding screenshot, forming a history of what the agent has attempted. The memory retains all textual action descriptions and plans while limiting screenshot retention to the most recent eight steps.

*Table 4.* Actions available at the top-level OS environment.

| Action | Description |
|---|---|
| open_app | Open an existing application, or open a file with an application. |
| close_window | Close a window |
| switch_window | Switch to a certain opened window based on window id |

We use gpt-5-medium (Singh et al., 2025) for planning agent and reflection agent, and UI-TARS-1.5-7B (Qin et al., 2025) for grounding.

## E. Case Study Details

### E.1. Case Study 1: Medical Secretary Task

**Task Description:** The agent acts as a lead medical secretary for an oncology testing center. Starting from a reference Excel spreadsheet containing patient information across multiple pathology labs, the agent must create three separate bulk forms (one per lab) and three corresponding email templates. Each spreadsheet must include the company logo, be filtered by lab, sorted by date, and contain additional columns with dropdown validation (Yes/No/N/A).

**Agent Trajectory:**

1. **[GUI] Open and examine the reference file.** The agent opens the original Excel file in LibreOffice Calc to visually inspect its structure, formatting, color scheme, and the embedded company logo.

2. **[Text] Analyze data structure with Python.** The agent uses openpyxl to programmatically parse the spreadsheet, identifying the header row, column mappings, unique lab names (Arizona, Canyon, Minnesota Pathology), and extracting style information (fonts, colors, borders).

3. **[Text] Generate three lab-specific spreadsheets.** The agent writes a Python script to create three new Excel files, each filtered by lab name, sorted by request date, with five new columns added and dropdown validation applied:

```
dv = DataValidation(type='list',
    formula1='"Yes,No,N/A"',
    allow_blank=True)
ws.add_data_validation(dv)
dv.add(f'I{data_start}:K{end_row}')
```

4. **[GUI] Verify generated file – discover missing logo.** The agent opens the newly created Arizona Pathol-

ogy spreadsheet and observes that while data and formatting are correct, the company logo (an embedded image) was not transferred by the Python script.

5. **[GUI] Copy logo from original file.** The agent switches to the original spreadsheet window, clicks on the REACH Oncology logo to select it, and copies it (Ctrl+C).

6. **[GUI] Paste logo to new file.** The agent switches to the Arizona Pathology window, navigates to the header area, and pastes the logo (Ctrl+V). This copy-paste operation is repeated for more than 2 times.

7. **[GUI] Verify dropdown validation.** The agent clicks on a cell in the "Order Received" column to confirm the dropdown menu appears with the expected "Yes", "No", "N/A" options.

8. **[Text] Generate email templates.** The agent uses `python-docx` to create three Word documents containing email templates for each lab.

**Key Insights:**

- **Modality selection based on capability:** The agent chose text-based Python for data manipulation (filtering, sorting, validation) but switched to GUI for embedded image transfer, recognizing the limitations of `openpyxl` with image objects.

- **Verification-driven workflow:** After programmatic generation, the agent proactively opened files via GUI to verify correctness, discovering the missing logo issue.

- **Efficient repetition:** The logo transfer required three iterations; the agent executed these systematically across all lab files.

### E.2. Case Study 2: Administrative Services Manager Task

**Task Description:** The agent acts as an Administrative Services Manager tasked with creating an FTE reduction package. The deliverables include: (1) a revised organizational chart reflecting a minimum 4% staff reduction, (2) an updated FTE report in Excel, and (3) a briefing note in Word describing the proposed reductions and their alignment with Budget Planning Principles.

**Agent Trajectory:**

1. **[GUI] Examine the organizational chart.** The agent opens the PDF containing the current org chart to visually understand the hierarchical structure and FTE counts for each position. This graphical document cannot be easily parsed programmatically.

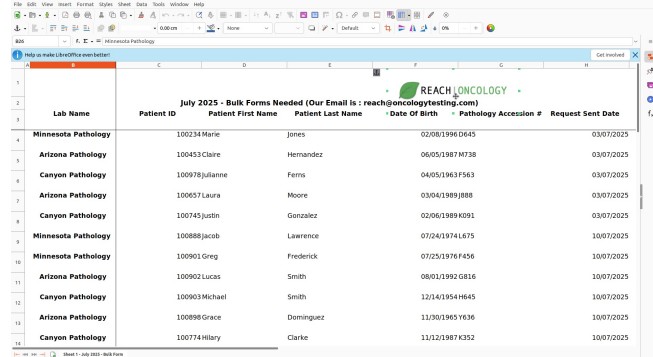

*(a)* Step 5: Agent selects the logo from original spreadsheet.

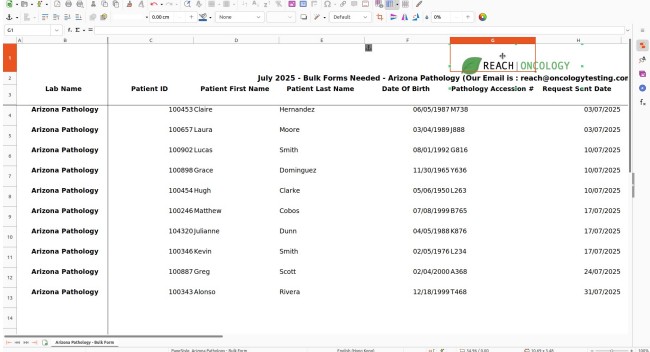

*(b)* Step 6: Logo pasted into new spreadsheet. Repeated 2 more times.

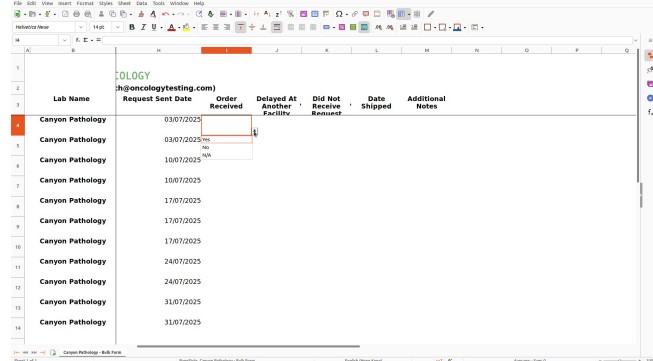

*(c)* Step 7: Verifying dropdown shows expected options.

*Figure 4.* Medical Secretary Task: Key GUI operations in the agent trajectory.

2. **[Text] Read budget principles and FTE report.** The agent uses `file_reader` to extract text from the Budget Planning Principles PDF (noting Principle #7 on strategic hiring), and uses `pandas` to parse the Excel FTE report, calculating baseline total of 237 FTEs.

3. **[Text] Plan the reductions.** Based on the task requirements (regional office reduction 10→9, voluntary attrition data), the agent calculates which positions to reduce: 6 FTEs from regional services (10% cut) and 7 FTEs from attrition, totaling 13 FTEs (5.5% reduc-

tion).

4. **[GUI] Attempt to edit chart in LibreOffice Impress – fails.** The agent converts the PDF to PNG and opens LibreOffice Impress to overlay edits. However, GUI click actions fail with internal errors.

5. **[Text] Adapt: Generate new chart with Graphviz.** The agent pivots to a text-based approach, writing Graphviz DOT code to recreate the org chart with updated counts and red highlighting for reduced positions:

```
rss_leads [label="Regional Support\
    nService Leads\n(9)\nChange:
    -1",
            color="#B00020",
                fontcolor="#B00020",
                    penwidth=2.2];
```

6. **[Text] Generate briefing note.** The agent uses `python-docx` to create a structured Word document with sections for purpose, background, proposed reductions, and explicit reference to Budget Planning Principle #7.

7. **[GUI] Verify deliverables.** The agent opens the generated PDF and Word document to confirm formatting and content accuracy.

**Key Insights:**

- **GUI for visual comprehension:** The agent used GUI to examine the graphical org chart, which contained spatial/hierarchical information difficult to extract programmatically.

- **Adaptive error recovery:** When GUI editing failed, the agent seamlessly pivoted to Graphviz code generation rather than abandoning the task, demonstrating robust fallback strategies.

- **Cross-document reasoning:** The agent synthesized information from multiple sources (org chart, FTE report, budget principles) to plan reductions and generate a coherent briefing note.

### E.3. Case Study 3: Maze Puzzle Task in Excel File

**Task Description:** The task requires the agent to solve a maze puzzle represented by colored cells in an Excel spreadsheet. Each cell in the attached spreadsheet represents a plot of land. The color of the cell indicates who owns that plot. Green cells are plots owned by Earl Smith. The Agent needs to determine if Earl can walk through every plot he

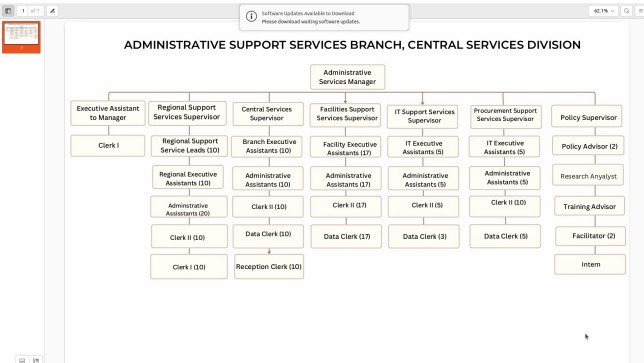

*(a)* Step 1: Examining the original organizational chart PDF.

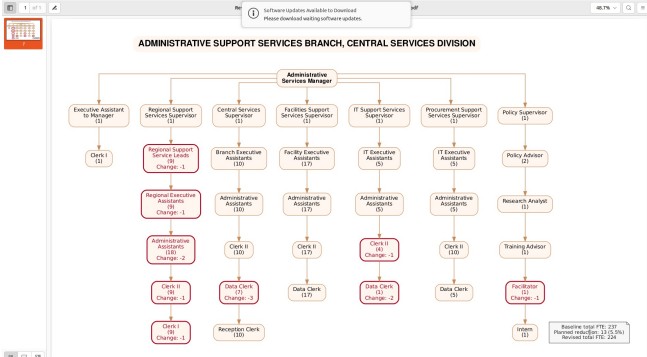

*(b)* Step 5: Revised chart generated via Graphviz with red highlights.

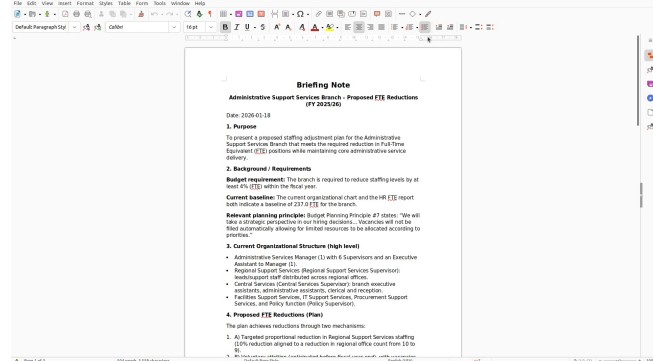

*(c)* Step 6: Briefing note generated via `python-docx`.

*Figure 5.* Administrative Services Manager Task: Key steps in the agent trajectory.

owns (and no other plots) and return to his starting plot without backtracking.

**Agent Trajectory:**

1. **[GUI] Open the Excel File.** With the GUI interface, the agent can simply open the GUI and figure out the answer.

**Key Insights:**

- **Unified interface provides more efficient agent solu-**

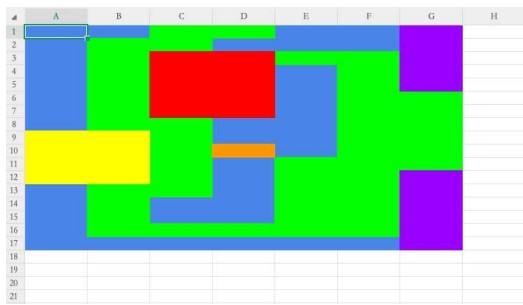

*Figure 6.* Maze Puzzle task: key steps in the agent trajectory.

**tions:** When solving the Maze Puzzle task with only text interface, the agent will first explore the excel file, try to re-build the excel and run a BFS to figure out the answer, which usually take more than 5 steps. However, with GUI interface, the agent can simply open the file and figure out the answer at step 1. The unified interface is providing a larger agent's action space that empowers the efficient solutions.

```
# BFS
seen=set(); stack=[coords[0]]
while stack:
    u=stack.pop();
    if u in seen: continue
    seen.add(u)
    stack.extend(adj[u])
print('connected',len(seen)==len(coords)
    )
# degrees summary
from collections import Counter
cnt=Counter(len(adj[u]) for u in coords)
print('degree counts',sorted(cnt.items()
    ))
# print grid representation
minr=min(r for r,c in coords); maxr=max(
    r for r,c in coords)
minc=min(c for r,c in coords); maxc=max(
    c for r,c in coords)
for r in range(minr,maxr+1):
    line=''
    for c in range(minc,maxc+1):
        line += 'G' if (r,c) in g else
            '.'
    print(line)
, timeout: 30, ')
```

