# OpenReview forum: "Position: Digital Agents Require Unified Agent-Native Environments"
_ICML.cc/2026/Position_Paper_Track — ICML 2026 Position Paper Track regular_

### Official Review · Reviewer_pEFs · 2026-03-03

**Significance:** 3
**Argument Clarity:** 3
**Rating:** 4
**Confidence:** 4

**Questions:**

Q1: How does the proposed AgentVM conceptually and practically differentiate itself from existing, highly capable local agent frameworks like OpenClaw? Given that OpenClaw already grants agents unified access to shell execution, browser control, and OS file systems, what specific novel capabilities does the agent-native computer paradigm unlock that these existing tools cannot?

**Alternative Views Section:**

Yes

**Compliance With Llm Reviewing Policy A Conservative:**

Affirmed.

**Discussion Potential:**

2

**Paper Summary:**

The authors strive to investigate the domain of environments for AI agents. The paper argues that current agent environments are highly fragmented, forcing agents to rely either on inefficient, error-prone GUIs or on narrow, task-specific text APIs. To address this, the authors propose the concept of an _Agent-Native Computer_. To ground this position, the authors introduce AgentVM, an implementation built on top of OSWorld. Through quantitative experiments on the OSWorld benchmark and qualitative case studies, the authors demonstrate that providing agents with unified access to GUI, OS-level text interfaces, and code execution monotonically improves performance and efficiency.

**Position:**

Yes

**Position In Title:**

Yes

**Related Work:**

3

**Strengths And Weaknesses:**

Pros:
* The paper provides a commonly accepted argument for why neither pure-GUI nor pure-text environments are sufficient for complex digital work. The case studies clearly illustrate the necessity of agent-friendly interaction environments.
* The ablation study on the OSWorld benchmark effectively supports the paper's core hypothesis, showing clear performance gains.
* The formalization of agentic environments as abstractions of the underlying POMDP provides a solid, structured way to think about agent-computer interfaces.

Cons:
* The paper positions the _unified agent environment_ as a missing, unsolved architectural gap that requires a novel conceptual framework like AgentVM. However, tools like OpenClaw already provide a good environment for agents, which is not aligned with the scope of the position paper. OpenClaw seamlessly blends full system access, browser control, and persistent memory directly on standard operating systems. The paper's claim that a purpose-built agent-native computer is strictly required feels slightly overstated when existing open-source agent frameworks are already achieving this unified system access in practice.

**Support:**

3

---

> ### Author Rebuttal · Authors · 2026-03-28
>
> ## Re: Weakness & Q1
> We thank the reviewer for raising OpenClaw. We agree that existing systems such as OpenClaw show that agents can already be built on top of standard operating systems with access to shell execution, browser control, file operations, and memory. **Our claim is therefore not that such agent runtimes do not exist, but that agents need a more explicit and principled environment layer over shared OS state.** More specifically, our contribution is an explicit intermediate layer between the agent and the OS: a unified action and observation space designed to expose system capabilities in a cleaner and more consistent way for agents. **A central goal of AgentVM is to investigate how to build a unified interface for both GUI-based and text-based interactions over the same underlying environment.** In this sense, the paper is less about arguing that practical agent runtimes are absent, and more about arguing that the interface exposed to agents remains under-specified as a general systems problem.
>
> Concretely, AgentVM is not just a collection of tools, but an environment abstraction that organizes interaction into coordinated views over shared OS state. This lets agents move naturally between text-based operations and GUI-based operations within one coherent interface, including desktop-native applications and OS-level surfaces as first-class interaction targets. For example, the same environment can support symbolic actions for structured manipulation, while also supporting GUI-based inspection, navigation, and verification when the task depends on rendered layout or native desktop interaction. **In this sense, our emphasis is on unifying desktop GUI interaction and text-based interaction within a single environment abstraction.** By comparison, OpenClaw is mostly focused on providing agent runtimes and tool access on top of the OS, with its documented GUI support being mostly browser-based. We will revise the paper to make this positioning clearer: the novelty we intend to emphasize is not the existence of OS-based agent runtimes, but the proposal of a more explicit interface layer for agents operating over a shared computer state.

---

> > ### Author Rebuttal · Reviewer_pEFs · 2026-04-01
> >
> > Thank you for the rebuttal. I have read it carefully and appreciate the clarification regarding the intended positioning of AgentVM.
> >
> > The final version should make this positioning much more explicit, especially in relation to existing systems such as OpenClaw, to avoid overstating novelty. I would like to raise my score to borderline accept.

---

### Official Review · Reviewer_fHY8 · 2026-03-12

**Significance:** 3
**Argument Clarity:** 2
**Rating:** 5
**Confidence:** 3

**Questions:**

- Is it possible that tasks that excel in one environment actually do worse when given this extra information as part of the agent-native setup?

**Alternative Views Section:**

Yes

**Compliance With Llm Reviewing Policy A Conservative:**

Affirmed.

**Discussion Potential:**

2

**Final Justification:**

I raised the score to accept and most main concerns are addressed. The argument clarity could be improved and it would be helpful for the authors to be very specific about the goals of the experiments.

**Paper Summary:**

The core argument of this paper is that digit agents should be evaluated in unified environments that allow for multiple types of interfaces including text, GUI and code. Existing environments usually operate in isolation e.g. GUI only or text only. The authors then propose a unified environment framework where multiple environments share actions, memory etc and provide some dataset results to backup the main claim.

**Position:**

Yes

**Position In Title:**

Yes

**Related Work:**

3

**Strengths And Weaknesses:**

Strengths:
- Addresses an important issue across benchmarking. People usually draw conclusions about methods based on the results but if external factors contribute to the results, it is hard to know what is going on.
- Provides logical arguments for why multiple environments together can be useful and connects with existing work
- Provides case studies where having a unified environment is helpful

Weaknesses
- The proposed agent-native framework relies a lot on screenshots. Screenshots are not natural images and aren to necessarily the best way to communicate information.
- The performance is still pretty low for the agent-native framework. This implies that there are other factors such as environment setup, action space etc that could contribute to discrepancies.
-  There is no discussion on how this might affect inference time and cost or potential difficulties in creating such an ideal agent-native framework.

**Support:**

3

---

> ### Author Rebuttal · Authors · 2026-03-28
>
> ## Re: Weakness 1
>
> We note that GUI interactions with screenshots and actions like click, type, etc, are standard practice for GUI agents[1][2][3], and we agree that screenshots are not the ideal communication medium for LLMs, and **this is precisely a motivation for our paper**.
> In Sec. 3, we explicitly argue that pure GUI interaction is inefficient for LLM agents because it forces a redundant text-to-image-to-text conversion loop. Our framework therefore does not advocate screenshot-only interaction; instead, it unifies GUI observations with structured text-based views over the same OS state. In Sec. 4, we further describe how application views can be augmented with richer symbolic observations and actions, **making agents less dependent on raw screenshots**. Our case studies in Sec. 5.3 shows that screenshots are mainly used for initialization and visual verification, while text/code tools perform the bulk of the task execution, and showcasing that screenshots are still necessary for agents to acquire visual information.
>
> ## Re: Weakness 2
>
> We agree that the absolute performance is still far from saturated. However, **our experimental goal is not to optimize a new agent to state-of-the-art performance, but to isolate the effect of the interaction interface itself**. Our main contribution is to study and explain for future digital tasks, and propose a conceptual framework of an unified surface that incorporates GUI-based and text-based interactions.
>
> Accordingly, we hold the agent structure fixed and vary only the interaction surfaces exposed by the environment. Under this controlled setup, performance improves monotonically from 41.54 to 51.30 as we progressively add tools, OS-level views, and code execution. We therefore view the results as evidence that unified interaction surfaces remove an important bottleneck, while not claiming that interface design alone is sufficient to close the full performance gap.
>
> We will revise the paper to state this more explicitly and to acknowledge that grounding quality, planning, action-space design, and environment robustness remain complementary factors.
>
>
> ## Re: Weakness 3
>
> We thank the reviewer for raising this point. Our framework can affect inference time and cost by reducing unnecessary GUI interaction. As discussed in Sec. 3.2, pure GUI use often requires long action chains, repeated screenshot parsing, and a redundant text -> image ->text loop, while structured text-based interfaces can expose the same state more directly and efficiently. In the Medical Secretary task, for example, the agent uses Python for the bulk spreadsheet transformation and uses GUI mainly for initial inspection and final visual verification, including checking formatting and restoring the logo when needed. When we tested with a pure GUI interface, the actions are taken so many steps that it can hardly complete the task.
>
> Creating an ideal agent-native computer does bring practical challenges. Building useful app-level views and actions takes engineering effort, and these interfaces may need updates as software and UI behaviors change. In addition, because the framework combines multiple views over the same OS state, the system must keep them synchronized and reliable. That said, we believe the conceptual framework remains important because it is compositional and modular: views can be added, grouped, and updated locally, making the environment easier to extend and maintain than a monolithic agent interface.
>
> Thank you for your valuable questions, and we will add these discussions in the appendix to cover these issues explicitly.
>
> ## Re: Question 1
> We agree this is possible in principle: if the agent cannot effectively arbitrate between modalities, additional interfaces may increase choice complexity and hurt some tasks. Empirically, however, our results do not show this pattern. Instead, we observe monotonic gains as the environment is enriched, which suggests that the added views are complementary rather than conflicting. The key point is that agent-native computers are not meant to force every task to use every modality, but to expose multiple modalities over the same OS state so the agent can use the one that is most efficient for the current subtask. Thus, what environments / tools to be provided to agents would be an important question to study as the action/observation space grows.
>
> [1] Xie, Tianbao, et al. "Osworld: Benchmarking multimodal agents for open-ended tasks in real computer environments." Advances in Neural Information Processing Systems 37 (2024): 52040-52094.
>
> [2] Agashe, Saaket, et al. "Agent s: An open agentic framework that uses computers like a human." arXiv preprint arXiv:2410.08164 (2024).
>
> [3] Li, Kaixin, et al. "Screenspot-pro: Gui grounding for professional high-resolution computer use." Proceedings of the 33rd ACM International Conference on Multimedia. 2025.

---

> > ### Author Rebuttal · Reviewer_fHY8 · 2026-04-03
> >
> > The authors mostly resolved my concerns. The one thing that I think should be included in terms of the screenshot discussion is that it should not be framed as an either-or e.g. screenshots or text and that an area for future work would be something that does not exist but can handle the limitations of both.

---

### Official Review · Reviewer_3qYB · 2026-03-12

**Significance:** 3
**Argument Clarity:** 3
**Rating:** 5
**Confidence:** 4

**Questions:**

1. In Tab 2, the addition of Tools and Code yields performance jumps. How much of this gain is derived strictly from the agent now having access to standard programmatic execution, v.s. the specific architectural novelty of the unified "Environment Views"?

**Alternative Views Section:**

Yes

**Compliance With Llm Reviewing Policy A Conservative:**

Affirmed.

**Discussion Potential:**

2

**Final Justification:**

Thanks for the rebuttal. I keep my positive rating and recommend acceptance of this paper.

**Paper Summary:**

Digital agents currently operate in fragmented environments, relying on either inefficient GUI interactions or limited text-based APIs. This paper proposes the "Agent-Native Computer," a unified environment integrating GUI and text interfaces over a shared operating system state. By structuring interactions into compositional Environment Views (e.g., OS-level, App-level), the framework aligns with LLM capabilities, allowing agents to seamlessly switch between programmatic execution for bulk tasks and GUI for visual verification. Implemented as AgentVM, the approach demonstrates monotonic performance gains on the OSWorld benchmark and solves complex tasks requiring multi-modal coordination along GUI automaton and CLI coding.

**Position:**

Yes

**Position In Title:**

Yes

**Related Work:**

3

**Strengths And Weaknesses:**

## Strengths
1. The transition from static LLMs to autonomous digital agents is a defining challenge of our current timeline; pointing out the fundamental mismatch between human-centric GUIs and LLM-centric action spaces is highly relevant and addresses a major bottleneck
2. Strong Conceptual Abstraction: The formulation of "Environment Views" (OS-level, App-level, Mounted Envs) provides a clean, principled, and compositional abstraction to merge heterogeneous interaction modalities.
3. Solid Empirical Grounding: Unlike many position papers that merely philosophize, the authors back their claims with a concrete implementation (AgentVM) and evaluate it using both quantitative OSWorld metrics and qualitative deep-dive case studies.

## Weaknesses
1. While the conceptual framing of "Agent-Native Computers" is compelling, the technical instantiation in the paper (AgentVM) heavily relies on gluing together existing frameworks. It combines the OSWorld environment , MCP tools (migrated from OSWorld-MCP) , and standard Python/bash script CLI execution. IMO, the paper needs to better defend why this is a fundamental architectural paradigm shift rather than just a 2026-era tool-use + CUA + Coding wrapper.
2. The quantitative experiment adds tools, an OS View, and Code execution progressively to a GUI-only baseline. However, it is unclear if the performance gains (+9.76% total) are strictly due to the unified dual-modality environment, or simply because the agent now has access to powerful programmatic tools. A "pure text-based" or "pure API" baseline on these exact workflow tasks is missing. Without it, we cannot isolate the delta provided specifically by the unification of modalities.

**Support:**

3

---

> ### Author Rebuttal · Authors · 2026-03-30
>
> We thank you for the positive assessment of our work's relevance, conceptual abstraction, and empirical grounding. We address your concerns and questions below.
>
> ### **W1. AgentVM Relies on Gluing Existing Frameworks**
>
> We agree that AgentVM integrates existing components, **and this is by design.** As stated in Section 5, **AgentVM is supporting evidence for our position, not a standalone systems contribution.** We believe that a position paper should validate its thesis with the simplest sufficient implementation, and the contribution lies in the conceptual framework, not in novel systems engineering.
>
> That said, **the integration is non-trivial in two respects.** First, **shared OS state** distinguishes our approach from simply bundling tools alongside a GUI agent. A file downloaded by a text tool is immediately visible in the GUI file manager, and a spreadsheet edited via Python can be verified through a screenshot. This bidirectional coherence requires careful design and is absent from naive tool-aggregation approaches. Second, the **Environment View abstraction** enables dynamic composition of interaction surfaces rather than hard-coding a fixed tool set. Views can be mounted and unmounted at runtime, allowing the environment to adapt to the task at hand.
>
> We will sharpen the paper's language to better distinguish conceptual and implementation contributions.
>
> ### **W2/Q. Unclear Whether Gains Come from Unification or Tool Access**
>
> We appreciate this question and would like to clarify the experimental design, since the layer-by-layer decomposition directly addresses this concern. **Programmatic execution is introduced only in the final layer. The first two layers add non-programmatic capabilities.**
>
> - **Tools** mounts structured text-based actions such as open_file as alternatives alongside GUI. These are not programmatic interfaces. They are action primitives exposed via MCP. The gain comes from replacing brittle multi-step GUI sequences with single structured calls over the same OS state.
>
> - **OS View** adds symbolic observations like window status and application info, along with OS-level actions like open_app and switch_window. No new task-completion capability is introduced here. The agent simply observes the environment through a complementary lens. This layer represents pure modality unification.
>
> - **Code** is the only layer where programmatic execution enters, providing batch processing and computation that neither GUI nor tool calls offer.
>
> We would like to note that **nearly two-thirds of the total gain comes from non-programmatic interface improvements.** This is directly attributable to the unified Environment View architecture. **Our case studies reinforce this.** In Case 1, Python handles data manipulation but the agent must switch to GUI to discover a missing logo. In Case 3, GUI renders the maze answer in one step versus 5+ steps of text-based spatial reconstruction. These gains arise from selecting the right modality per subtask, which is the core capability a unified environment enables.
>
> Regarding your suggestion on adding baselines in W2, we would like to note that **a pure text-only baseline cannot complete many of the OSWorld tasks** because they require visual verification and screen operations, making the implementation of a text-only baseline infeasible.

---

> > ### Author Rebuttal · Reviewer_3qYB · 2026-04-03
> >
> > I recommend acceptance of this paper.

---

### Official Review · Reviewer_BBn6 · 2026-03-16

**Significance:** 3
**Argument Clarity:** 3
**Rating:** 4
**Confidence:** 4

**Questions:**

Questions

  1. What specific base model was used in the OSWorld experiments? Are the incremental gains from each interface layer consistent across different models (e.g., GPT-4o vs. Claude 3.5 vs.
  open-source VLMs)?
  2. How many tasks are in the OSWorld workflow subset? Were the results in Table 2 averaged over multiple runs? What are the standard deviations?
  3. Do you have data on average token consumption or inference cost across different settings? What is the tradeoff between performance gains and cost increases?
  4. For primarily text-based tasks (e.g., SWE-bench-style coding tasks), does the unified interface still provide improvements? Or is this position mainly applicable to the subset of tasks
   requiring visual interaction?
  5. The Environment View design is currently hand-crafted (e.g., OS-level actions in Table 4 are predefined). Have you considered allowing agents to automatically discover or construct
  views? What are the implications for scalability?

**Alternative Views Section:**

Yes

**Compliance With Llm Reviewing Policy A Conservative:**

Affirmed.

**Discussion Potential:**

3

**Paper Summary:**

This paper argues that digital agents need a unified Agent-Native Computer rather than the current fragmented, task-specific interaction environments. The core claim is that real-world
  digital tasks simultaneously require GUI interaction (visual inspection, formatting verification) and text-based interaction (bulk data processing, code execution), and no single modality
   suffices.

  The paper proposes a conceptual framework centered on Environment Views—different (observation, action) factorizations of the underlying OS state, including OS-level views, App-level
  views, mounted environments, and free-form actions. Based on this framework, the authors implement AgentVM (built on OSWorld) and progressively add Tools, OS View, and Code layers on the
  OSWorld workflow subset, demonstrating monotonic performance improvements (41.54 → 51.30). Three case studies from GDPVal and GAIA qualitatively illustrate the advantages of a unified
  interface.

**Position:**

Yes

**Position In Title:**

Yes

**Related Work:**

3

**Strengths And Weaknesses:**

Strengths:

  1. Timely and important topic. The fragmentation of agent environments is a real pain point in current LLM agent research. The paper clearly articulates the limitations of existing
  environments—Table 1's comparison of benchmark capabilities is particularly effective—and identifies the inefficiency of GUI-only approaches and the coverage gaps of text-only approaches.
   This has practical guiding value for the community.
  2. Clean conceptual framework. The Environment View abstraction—decomposing environment interaction into multiple (O, A) views over shared OS state—is intuitive yet provides a clear
  mental model for organizing agent interaction surfaces. The design principles that all views share the same underlying OS state and that views can be dynamically mounted/unmounted are
  useful for building flexible agentic systems.
  3. Well-designed ablation on OSWorld. Fixing the agent architecture and progressively layering interface components is a clean experimental design. The monotonic improvement in Table 2
  effectively supports the claim that unified interfaces outperform single-modality ones. The largest increment from +Tools (+5.18 points) directly demonstrates that text-based tools can
  replace lengthy GUI action chains.
  4. Well-chosen and persuasive case studies. The Medical Secretary task—where the agent uses Python for bulk data processing then switches to GUI to discover and fix a missing logo—and the
   Maze Puzzle—where GUI immediately reveals the answer vs. text requiring spatial layout reconstruction over 5+ steps—are compelling illustrations of why a unified interface is necessary.
  5. Comprehensive related work. Appendix C's categorization of agentic benchmarks (Web, Enterprise, Computer Use, Mobile, Deep Research) is thorough and serves as a useful reference for
  the community.

  Weaknesses:

  1. Thin quantitative evidence with incomplete reporting. This is the most significant issue.
    - Quantitative experiments are limited to a single benchmark (OSWorld workflow subset). The number of tasks in this subset is not reported, nor are confidence intervals or statistical
  significance.
    - The base model used is never specified—the OSWorld experiments only state the agent architecture is inspired by Agent S2.5, but the specific LLM is never mentioned. This is a major
  omission, as GUI grounding capabilities vary dramatically across models, leaving the generalizability of the conclusions uncertain.
    - The case studies use GPT-5.2 + UI-TARS-1.5-7B (Appendix D), potentially a different model from the OSWorld experiments, creating inconsistency across experimental settings.
  2. Limited novelty of the position itself. "Agents need both GUI and text interfaces" is a relatively obvious conclusion to practitioners—commercial products like Claude Cowork, Manus,
  and Eigent already validate this in practice (the paper itself cites these). The POMDP formalization and Environment View definitions add formality but yield limited new insight.
  Definitions 1–3 essentially restate in symbols the intuition that "different interfaces expose different observations and actions."
  3. No discussion of efficiency/cost tradeoffs. More views mean longer context (OS state, screenshots, tool descriptions all enter the prompt), more API calls, and a more complex agent
  decision space. The paper does not report token consumption, inference time, or cost across different settings. If the unified interface's performance gains come with doubled cost, the
  implications for practical deployment need to be discussed.
  4. Alternative Views discussion lacks depth. Section 6's response to "APIs alone suffice" (expanded in Appendix A) mainly argues that proprietary apps lack programmatic interfaces,
  accessibility trees are incomplete, and visual verification is needed. But it does not directly address: in the currently most successful agent application scenarios (SWE-bench, coding
  agents), pure text is indeed sufficient. In other words, the paper's position may hold more strongly for specific task categories (office work, visual tasks) than as a universal
  conclusion—this nuance should be acknowledged.
  5. Ambiguous positioning of AgentVM's contribution. AgentVM is essentially OSWorld + MCP tools + OS-level actions + code execution. As supporting evidence for a position paper this is
  appropriate, but the paper's language sometimes implies it is an independent systems contribution. Implementation details in Appendix D are also quite brief.

**Support:**

3

---

> ### Author Rebuttal · Authors · 2026-03-30
>
> We appreciate your recognition of our paper's timeliness, clean conceptual framework, well-designed ablation, persuasive case studies, and comprehensive related work. We address your concerns and questions below.
>
> ## Weaknesses
>
> ### **W1. Base Model, Task Count, and Statistical Reporting**
>
> As noted in Appendix D, we use **gpt-5-medium** for planning and **UI-TARS-1.5-7B** for grounding, following recommended setup from Agent S2. The same configuration is used throughout OSWorld experiments. We will state this clearly in the main text.
>
> The workflow subset contains **101 tasks**, the most challenging tier in OSWorld. Each task requires cross-application navigation, and all prior works report their lowest scores here. We conducted single runs, consistent with prior OSWorld evaluations, none of which report confidence intervals. We clarify this in the main text.
>
> Regarding generalizability: the ablation holds the model fixed, so gains are attributable to the interface layers. Each layer provides structurally different capabilities. These structural advantages are model-independent, so we expect the direction of gains to hold across capable VLMs, though absolute magnitudes will vary.
>
> ### **W2. Limited Novelty of the Position**
> Our paper does not merely observe that "agents benefit from multiple modalities." **It formally proposes and advocates that generalist agents *require* both GUI and text-based interaction surfaces.** While products like Claude Cowork and Manus offer multi-modality features, none provide a principled framework for *why* and *how* interaction surfaces should be composed. Specifically, we (1) formalize the problem through the Environment View abstraction, telling practitioners *what to build* and *how to evaluate*; (2) conduct controlled experiments identifying *when* each modality helps; and (3) identify design principles that distinguish a coherent environment from ad hoc tool bundling. No prior work unifies these surfaces formally and validates the framework empirically.
>
> **Recent industry trends further exemplify our position.** Anthropic's Claude Code supports a multi-tier interaction model combining connectors, browser control, and screen control over shared state. Daytona raised $24M in February 2026 on the thesis that agents need composable computers with stateful sandboxes. These developments independently converge on the architecture we formalize. We also provide case studies in Section 5.3 showing concrete scenarios where agents must switch between GUI and text modalities to complete real-world tasks, demonstrating the necessity of unification.
>
> ### **W3. No Discussion of Tradeoffs**
>
> The unified interface is actually *more efficient*. We report average API calls, wall-clock time, and task success rate below:
>
> | Setting | Avg API Calls | Avg Time (min) | Result |
> |---------|---------------|----------------|--------|
> | Baseline | 75.81 | 22.66 | 41.54 |
> | + Tools | 72.13 | 24.23 | 46.72 |
> | + Tools + OS View | 81.52 | 25.67 | 47.87 |
> | + Tools + OS View + Code | 68.77 | 23.47 | 51.30 |
>
> The full configuration uses *fewer* API calls and comparable time versus the GUI-only baseline and achieves better results. Text tools replace multi-step GUI chains with single calls; code execution batches operations that would require many sequential GUI actions. We will add this to the revision.
>
> ### **W4. Position May Not Hold Beyond Visual/GUI Tasks**
> We agree that purely text-based tasks like SWE-bench do not require GUI, as discussed in Section 6.2. Our position addresses a broader question: **what should the environment look like if it must support any digital task?** We believe that **coding is one slice of the broader agent opportunity.** McKinsey's 2025 Global Survey reports 62% of organizations are experimenting with AI agents across healthcare, finance, operations, etc. GDPVal evaluates agents across 44 occupations in 9 sectors. These domains overwhelmingly involve GUI-based software lacking comprehensive APIs.
>
> ### **W5. On Positioning of AgentVM's Contribution**
> We agree. AgentVM is supporting evidence for our position, not a standalone systems contribution. We will revise the language to reflect this distinction and add more implementation details.
>
> ## Questions
> Q1, Q2, and Q3 are addressed in W1 and W3 above. Q4 is addressed in W4.
>
> ### **Q5. Automatic View Discovery**
> In this paper, **environment Views are hand-crafted by design to validate the conceptual framework.** Automatic discovery is a natural extension: the MCP ecosystem supports dynamic tool registration, and our view mounting mechanism was designed with this in mind. Recent work on tool forging[1] shows LLMs can construct tools from experience, suggesting a path toward autonomous view discovery. We will discuss this in the revision.
>
> ## Reference
> [1] Qiu et al. Alita: Generalist agent enabling scalable agentic reasoning with minimal predefinition and maximal self-evolution.

---

### Decision · Program_Chairs · 2026-04-30

**Decision:**

Accept (regular)

**Comment:**

The paper makes a timely and well-motivated case that unified environment design is an important missing piece for general-purpose digital agents, and the overall review is positive. Remaining weaknesses mostly concern scope, reporting detail, and cost-efficiency tradeoffs, but these feel addressable in revision rather than fatal to the core case. I believe the paper is strong enough for acceptance as a position paper.